# Optimizations of Double Titanium Nitride Thermo-Optic Phase-Shifter Heaters Using SOI Technology

**DOI:** 10.3390/s23208587

**Published:** 2023-10-19

**Authors:** Eylon Eliyahu Krause, Dror Malka

**Affiliations:** Faculty of Engineering, Holon Institute of Technology (HIT), Holon 5810201, Israel; eylon1909@gmail.com

**Keywords:** titanium nitride, SOI, silicon, phase shifter, MZM

## Abstract

A commercial thermo-optic phase shifter (TOPS) is an efficient solution to the imbalance problem in the fabrication process of Mach–Zehnder modulator (MZM) arms. The TOPS consumes electrical power and transforms it into thermal energy, which changes the real part of the effective refractive index at the waveguide and adjusts the MZM transfer function to work in the linear region. The common model being used today is constructed with only one heater; however, this solution requires more electrical power, which can increase the transmitter system cost. To reduce the system energy cost, we propose a pioneering optimal double titanium nitride heater model under forward biasing at 1550 nm wavelength using the standard silicon-on-insulator technology. Numerical investigations were carried out on the key relative geometrical parameters, heat distribution at the silicon layer, thermal crosstalk, and laser wavelength drift. Results show that the optimal TOPS design can function with a low electrical power of 19.1 mW to achieve a π-phase shift, with a low thermal crosstalk of 0.404 and very low optical losses over 1 mm length. Thus, the proposed device can be used for improving the imbalance problem in MZMs with low electrical power consumption and low losses. This functionality can be utilized to obtain better performances in transmitter systems for data centers and long-range optical communication system applications.

## 1. Introduction

The integration of thermo-optic phase shifters (TOPSs) using silicon-on-insulator (SOI) technology presents an appealing solution for various applications, including sensors, high-speed communications, filters, and broadband electro-optic modulators. In these applications, TOPSs play a crucial role in encoding light through intensity or phase modulation [1,2,3,4,5]. TOPSs are being created monolithically, usually with silicon (Si) as the waveguide (wg), other dioxides semiconductors like silica (SiO_2_) as the covering material, and other metals/doping semiconductors for the heaters. TOPSs are being integrated into a carrier depletion-based Mach–Zehnder modulator (MZM), a device that modulates a signal by either difference in length or a change in the effective refractive index of the transverse electric (TE) mode due to heating [2]. This phenomenon arises from the depletion carrier effect or thermal changes around the MZM. Heating charges create differences in voltage potentials, which start a charge movement inside the MZM and consequently change the operation voltage [6,7]. The relation between the operating voltage and the optical signal at the output is co-sinusoidal. The linear area is defined as an approximation around the bias quadrature points, which occurs firstly at π/2. TOPSs heat the MZM in a uniform pattern and thus keep the operating voltage at the linear region, which allows us to transmit signals with a bandwidth as wide as 25 Gbit/s and beyond, and maximize the signal-to-noise ratio [8,9,10,11]. The TOPS requirement comes from the imprecise fabrication of the MZM; the two lines that split the optical signal, in which the signal propagates through each one of them and then combines the two for phase modulation, are not identical in length, and a difference in their lengths causes a difference in the phase from each path. A phase difference, unless intentionally applied (like the TOPS itself), needs to be avoided. Otherwise, it would create non-in-phase interference at the end of the MZM, which would damage the signal, and in addition, in an integrated feedback loop, it could cause destabilization with positive feedback [12,13,14,15]. TOPSs are an efficient solution for the imbalance problem [4], yet there has not been a use case for more than one heater in TOPSs. This paper commences with the optimization of a dual metal-based titanium nitride (TiN) TOPS design with thermal simulation based on finite difference equation (FDE) and optical simulations based on the finite element method (FEM). The metal-based heaters were chosen due to the TiN advantage over doped semiconductors in attributes such as low specific heat (598 J/kgK) versus (711 J/kgK) in Si, while having roughly the same thermal conductivity (28 W/mK) as Si based phosphorus doping (25 W/mK) [4]. Its separation from the wg has negligible attenuation coefficient, making it a superior solution for long range communication systems. The simulation in the Lumerical suite was performed to the thermal crosstalk between the two heaters of an MZI switch for both TOPS types, for various heater powers, and for various distances between the heaters. This paper thus comprehensively reports on double TOPS design optimizations and crosstalk mitigation in a standard SOI process, which, to the best of our knowledge, had not been presented in the literature [16,17] and can be an efficient solution with low power consumption for optical communication systems.

## 2. Design of the Device and Theoretical Aspects

Figure 1 shows the Si strip waveguide structure in the x-y plane, while the light propagates on the z-axis. The colors in the figure are defined as follows: yellow is air, light blue is the TiN heaters, light grey is SiO_2_, and purple is Si wg. The dimensionality of the components are as follows: the height of the air is 5 μm; the height of the SiO_2_ (h_SiO_2__) is 2 μm, which also called buried oxide (BOX); the height of the Si wg (h_wg_) is 220 nm, and its width (w_wg_) is 500 nm. The dimensionality is required to obtain a fundamental TE mode solution to the operated wavelength (λ_0_) of 1550 nm (which can be used in TM mode as well). The device length (along the z axis) is 1 mm. The metal used in the double heaters is TiN, and their locations in the structure were optimized to obtain better performances; the optimization parameters are the distance between the two heaters (d_TiN_) as well as their center of mass distance from the Si wg (d). The TiN dimensions were set to be suitable to the fab capability fabrication conditions that exist today. Hence, each TiN heater has the same height of 220 nm (h_TiN_) and width of 2 μm (w_TiN_).

In the following study case, there is an optimization of a model that uses a dual heater structure based on TiN. The heating units keep the temperature constant across the model, which changes the electron distribution and approximately keeps the biased voltage in the linear region. The crucial input, aside from the materials and geometrical aspects, is the electrical power. To validate and use the model with a driven voltage, the relation between Ohm’s law, the electrical power of the resistor, and the formula for solid resistance will yield the following:(1)$$V_{\pi }=\sqrt{P_{\pi }\frac{A}{\sigma l}}$$
where P_π_ is the total power needed for a π-phase shift, σ is the electrical conductivity, A is the surface area of the resistor, and l is the length of the resistor. Since the geometrical and material aspects are constant, we can see a square root relation between the operating voltage and the total power. In the simulation, we utilize the Heat tool, a simulation for studying semiconductor devices based on a discrete approximation of thermodynamics and its relation to Maxwell’s equation. The results are both graphic, by heat distribution plotting, and numerical, relating the consumed electrical power to a coordinate on the media. The heat transport equation is given by:(2)$$\rho c_{\rho }\frac{\partial T}{\partial t}-\nabla (k\nabla T)=Q$$
where T denotes the temperature at each media coordinate, ρ represents mass density, c_ρ_ denotes specific heat, k stands for thermal conductivity, and Q represents the rate of applied heat energy transfer. By incorporating the electrical conductivity σ, the electric current equation (Ohm’s law) can be concurrently solved with the heat transport equation. Internally, in the media, the charge density may change its distribution at each point, over time, due to the external voltage applied. By plugging the continuity equation, we will be able to determine the charge density both in the media space and its change over time.
(3)$$\frac{\partial \rho }{\partial t}=-\nabla J$$

The Poisson equation describes the relation between charge distribution and the external voltage in the media. To avoid unnecessary oscillatory behavior in the distribution, we solve for the DC steady-state power consumption:(4)−∇(ε∇V)=ρ

The DC step response, which contributes to heat dissipation, is stable since it generates an exponential decay in the charge density, which corresponds to a left pole.
(5)$$\frac{\partial \rho }{\partial t}+\frac{\sigma }{\varepsilon }\rho =0$$

The decay process is defined by the relaxation time τ = ε/σ. When examining changes in the system over time, scales much greater than τ (t >> τ), such as the quasi-static approximation, becomes relevant, resulting in a negligible value for ∂ρ/∂t. It should be emphasized that in steady-state conditions, the rate of change of the charge density (∂ρ/∂t) is intentionally disregarded. In the case that the charge density changes much slower for the relaxation time, the solution becomes much simpler by being homogenous [18,19,20,21]: (6)∇(σE)=0

Simply by using Ohm’s law once again, we can directly relate the emerging change in the media’s electric field and the given electrical power consumption:(7)$$P=JE=\sigma E^{2}$$

The dissipated power resulting from Ohmic loss is accounted for in the heat transport equations as the transfer rate of heat energy, where the entire power consumption is converted into heat. The HEAT tool discretizes and solves either the heat transport equation alone or the coupled heat transport and electric current equations using a finite-element mesh on a media’s surface or throughout its volume. The simulation region is divided into multiple domains along material boundaries, each possessing distinct physical characteristics. The accuracy of the simulation can be changed by the number of points in the media, that are being taken into account when solving the equations. Accuracy itself is a tradeoff with the runtime of the simulation. The materials utilized in the simulation range from insulators to conductors and even fluids. Each material type has its own parameters; for some, they are already given, and for others, they can be implemented or changed by the user. The system of equations addressed in a heat transport simulation accommodates both steady-state and time-varying solutions. Since the simulation works in both ways, now, we can set initial conditions for the temperature and have a constraint to keep it constant over time: (8)$$\frac{\partial T}{\partial t}=0$$

We will now have to solve a special case for the heat equation with its restriction:(9)−∇(k∇T)=Q

Steady-state simulations allow for the examination of the system’s performance under a consistent operating condition. In contrast, by specifying an initial temperature (and electric field if required), the equations can be solved at desired time periods. This enables the evaluation of the component’s time-dependent behavior, facilitating a direct analysis of its response in the time domain, particularly for signals with long-time oscillation analysis [22,23,24,25,26,27,28]. 

Increasing the voltage show a respective increase in the Ohmic heating. The factor that needs to be taken into account for heating is thermal crosstalk, which is defined as the ratio between the increment in temperature at the passive component and the active component. In our case, the passive component is the Si wg, and the active is the double TiN heaters pair above the wg [29]. The cross talk is a good approximation to its analog measurement given that in the simulation, we view the crosstalk on the surface of XY plane, as seen in Figure 1, as a matrix, with each point being summed over is a location on the matrix:(10)$$\mathrm{Normalized}-\mathrm{thermal}-\mathrm{crosstalk}=\frac{\sum_{n,m}Y_{\mathrm{nm}}-Y_{\mathrm{ambient}}}{\sum_{q,r}X_{\mathrm{qr}}-Y_{\mathrm{ambient}}}$$
where Y_nm_ is a sample of passive components surface at location n,m; X_qr_ is a sample of active components surface at location q,r; and Y_ambient_ = X_ambient_ = 300 K is the TOPS at ambient temperature. The ohmic heating itself, as mentioned above, is causing a phase shift, which we are using to stabilize the imbalance problem. The temperature-dependent phase change in the heated waveguide is:(11)$$\Delta \phi =\frac{2\pi L}{\lambda_{0}}\frac{\mathrm{dn}}{\mathrm{dT}}\Delta T$$
where L is the phase shifter length, λ_0_ is the operated wavelength of 1550 nm with a Si thermo-optic coefficient of dn/dT~1.8 × 10^−4^ K^−1^, T = 300 K, used as the room temperature and the respective temperature change ΔT. The temperature difference limit will occur at our maximum phase shift (Δϕ = π):(12)$$\Delta T_{\pi }=\frac{\lambda_{0}}{2\mathrm{Ldn}/\mathrm{dT}}$$

It is crucial to consider that the device length scale typically ranges in the hundreds of micrometers. This is because the resistance of the heaters increases proportionally with their length. Consequently, higher voltages would be required to drive the system, leading to reduced power efficiency. Additionally, the phase-shift calibration should be limited to a maximum of π-phase shift. Beyond this point, there is no significant advantage gained in terms of additional phase shifting; rather, it would result in unnecessary additional power consumption [4]. The heat dissipation is dependent on the attributes of the active and passive components. The time constant of the TOPS (τ) is defined by the amount of time for a change of 1 × 10^−1^ from the total change in the steady state and is given by:(13)$$\tau =\frac{H}{GA}$$
where G denotes the thermal conductance between the heated Si wg and the Si substrate, and H denotes the heat capacity of the TOPS. 

The figure of merit (FOM), which is the total power consumption multiplied by the heat dissipation, is given by:(14)$$\mathrm{FOM}=P_{\pi }\tau$$

To examine our TOPS efficiency and to select the optimal design, a calculation of the FOM has been performed to evaluate the TOPS performances.

## 3. Simulation Results

The TOPS was analyzed and solved using HEAT solver and MODE tools based on Lumerical suite software. The optical simulations were performed using the MODE solver tool, which is based on the finite difference eigenmode (FDE) numerical method. The heat simulations were performed using the HEAT solver tool, which solves the system of equations describing the distribution temperature as a function of power. In our simulations, all height and width values are taken into the constraints of the fab fabrication ability, where the smallest width of the heaters allowed after fabrication is 2 μm. The heat diffusion equation in an area of 8 μm width × 3.5 μm height. Key parameters such as thermal conductivity are taken into account. 

Figure 2a–d shows the thermal profile of the TOPS with different distance between the two TiN heaters. The high peek temperature of the TiN heaters (339–336 K) can be noticed in Figure 2a–d, and the heaters location was set above 1.1 μm in order to avoid the possibility of light coupling from the Si wg to the heaters, which is a key feature to obtain the very low attenuation coefficient.

After obtaining the index delta data from the HEAT solver, the fundamental mode of the Si wg-based TOPS is being solved by mode solver for single TE mode at 1550 nm wavelength. Figure 3a–e shows the fundamental mode solution profile of the double TiN heaters at different distances between the heaters, and from this solution, the change of the effective index can be obtained for each design. Thus, combined with Equation (1), the voltage for obtaining phase change of π can be found, as shown in Figure 3a–e.

Figure 4 shows the phase shift as a function of the total heaters power for five different distances between the heaters. It can be noticed from Figure 4 that more electrical power for obtaining a phase shift of π is needed in cases of the distance between the two heaters is over 1 µm. Hence, the study of the tradeoff between the electrical power and the thermal crosstalk needs to be investigated to select the optimal design.

To choose the optimal design, a lower thermal crosstalk must be obtained, such that the contribution of one heater unit would not overheat the other unit and affect the heating of the Si wg area. Overheating can cause unwanted phase-shift deviation, which can easily lead to more optical losses [30,31,32,33,34,35,36,37]. Figure 5 shows the thermal crosstalk versus the distance between the two heaters. The results from Figure 2, Figure 3, Figure 4 and Figure 5 have been analyzed in order to select the optimal design. From Figure 5, it can be observed that a higher normalized thermal crosstalk over 0.45 is obtained in cases of short distance between the heaters which is less than 0.85 µm. Thus, the best design must have less than 0.45 normalized thermal crosstalk. Moreover, the shorter the distance between the heaters, the more efficient the power consumption will be, which means that the consumption will be lower to obtain a π-phase shift, but at the same time, the tradeoff with thermal crosstalk will be greater. Therefore, the optimal design consists of a 1 μm distance between the two heaters. In this case, the power for obtaining a π-phase shift is considerably lower, with 18.95 mW total power and a low normalized thermal crosstalk of 0.404. 

Another important characteristic is the height distance between the double TiN heaters and the Si wg. The TiN heaters have a positive imaginary refractive index that can increase the optical losses in case the heaters are located close to the Si wg. In addition, in this case, less electrical power is needed to obtain a phase shift of π. Thus, an optimization on the d parameter is needed to be achieved in order to understand the tradeoff between optical losses and power consumption. Figure 6a–d shows the thermal profile of the TOPS with a different d value (0.5–2 μm). The resultant data were analyzed by solving the TE mode and extracting the power for the phase shift of π and the absorption coefficient (attenuation). In these figures, it can be noticed that a strong thermal change over the Si wg area is obtained at short distances of d, as shown in Figure 6a,b.

Figure 7 and Figure 8 shows the optimization of the d value, which includes how much power is needed to obtain the phase shift of π at different values of d and the attenuation (losses) for each case. From the results shown in Figure 7 and Figure 8, it is clear that the optimal d value is 1 µm, with a low power of 19.1 mW and without suffering losses. Thus, selecting this distance ensure that the light of the TE fundamental mode does not has an overlap with the double TiN areas. 

Figure 9 shows the tolerance sensitive of the total π-phase shift power consumption for the wavelength drift effect over the C-band spectrum. This investigation is crucial because the wavelength in which the laser operates is not constant and has a shift-drift effect around the desired wavelength due to the heating process of the laser lifetime. Hence, a shift of around 0.1 nm per degree Celsius can happen for Si wg [38], and usually, a commercial laser can cause a heat increase of over 50 degrees from room temperature. It can be observed from Figure 9 that the proposed TOPS has a good sensitive to the laser drift effect, with only a minor power shift of 60 μW due to a temperature increase of 50 Celsius degrees due to laser heating and an overall good power consumption stability over the C-band spectrum, with only a small power shift of ±0.25 mW from the operating wavelength.

The rise time of the optimal TOPS design was found to be 2.31 μs, which is the time it takes the heat to dissipate (heat diffusion). This time constant can be theoretically improved by fabricating TiNs with a shorter width. In our case, the width was selected to be 2 μm because of the fabrication limit. To understand the advances of this new design, a comparison between the main properties of the proposed TOPS and other state-of-the-art TOPS components was performed, where all TOPS are operated under forward bias operation mode. The main properties are TOPS type, TOPS length (L), optical losses from the heaters, voltage for π-phase shift (Vπ), power for π-phase shift (Pπ), rise time (τ), FOM, and the year of publication. The results in Table 1 show that our double TiN heater design is superior to other TOPS designs in all aspects. Our design has the best power electric consumption, 19.1 mW, with very low optical loss due to the heaters and fast switch time (rise time) using a PS with a length of 1 mm. Another advance of this technology compared to the doped Si TOPS design is the ability to design a PS with a large length without consuming too much electrical power. Thus, for designing an MZM with a large RF line, it will be better to use a double TiN heater design to reduce energy costs.

## 4. Conclusions

A new TOPS design based on double TiN heaters using a commercial 220 nm SOI strip wg under forward biased has been studied in this research. This study shows how to design TOPS with a high energy efficiency that can be suitable to fulfill the commercial requirements for designing MZMs that work with RF lines around 1 mm in length. The optimal parameters of the TOPS design are a distance between the two TiN heating units of 1 μm, a distance between the center of the heaters mass and the upper surface of the Si strip wg of 1 μm, and a PS length of 1 mm. The new TOPS design shows good stability to the thermal laser drift effect, with only a minor power shifting of ±0.25 mW across the C-band spectrum. The results show that the proposed TOPS can be used for shifting a π-phase shift using low electrical power (19.1 mW) over 1 mm PS length, with an excellent normalized thermal crosstalk of 0.40 between the two TiN heater units, a fast rise time of 2.31 μs, very low optical losses of 7.72 × 10^−9^ dB/mm, and FOM of 44.12 mWμs. Thus, the new TOPS design grants us groundbreaking energy efficiency compared to other TOPS designs, such as doped Si or only using one TiN heater unit. This new TOPS can be easily integrated into MZMs that work under high-speed conditions, which usually require a large RF line (around 1 mm) to solve the imbalance arms issue, with a low cost of electrical energy. From a commercial view, this new design is very attractive because of the ability to reduce the driver voltage level for achieving phase shift control in the MZM. Thus, this ability from a system view can be utilized for reducing the transmitter costs.

## Figures and Tables

**Figure 1 sensors-23-08587-f001:**
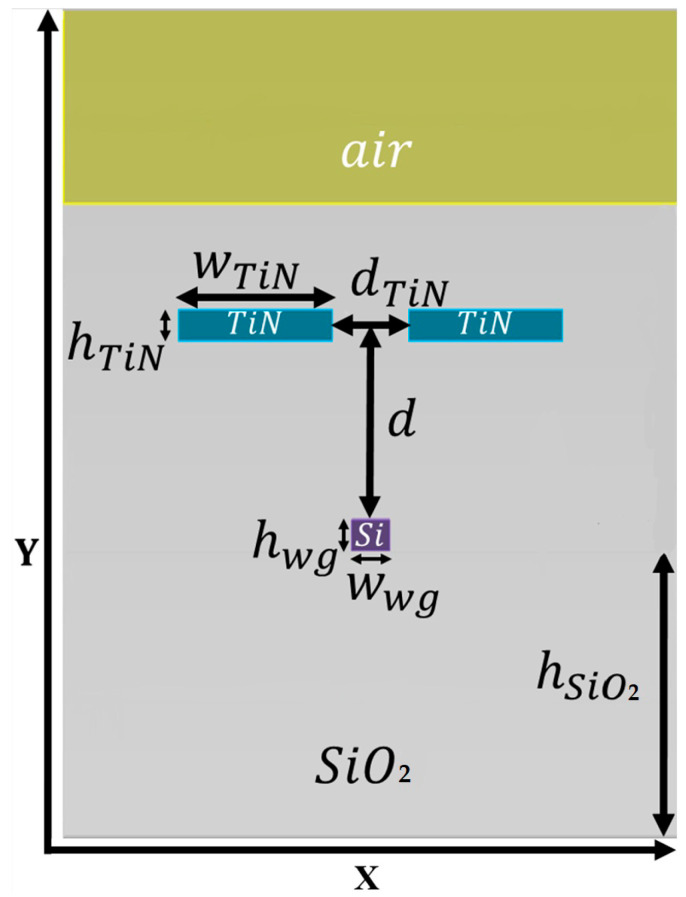
Schematic sketch of double TiN in Si strip wg structure in the x–y plane.

**Figure 2 sensors-23-08587-f002:**
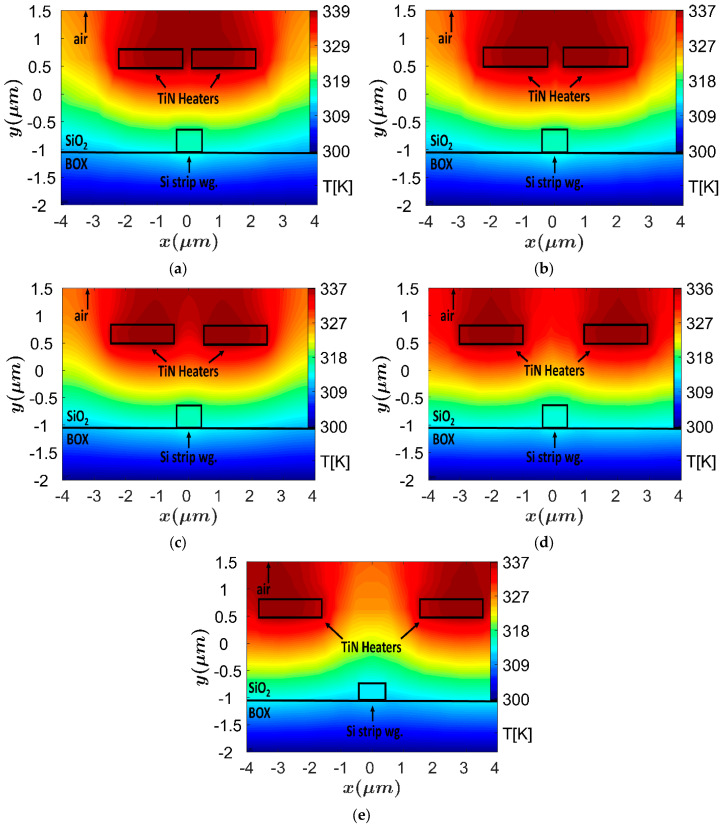
The heat distribution at the XY plane, each plot represents the change in the distance between the heaters: (**a**) d_TiN_ = 0.5 μm, (**b**) d_TiN_ = 0.75 μm, (**c**) d_TiN_ = 1 μm, (**d**) d_TiN_ = 2 μm, (**e**) d_TiN_ = 3 μm.

**Figure 3 sensors-23-08587-f003:**
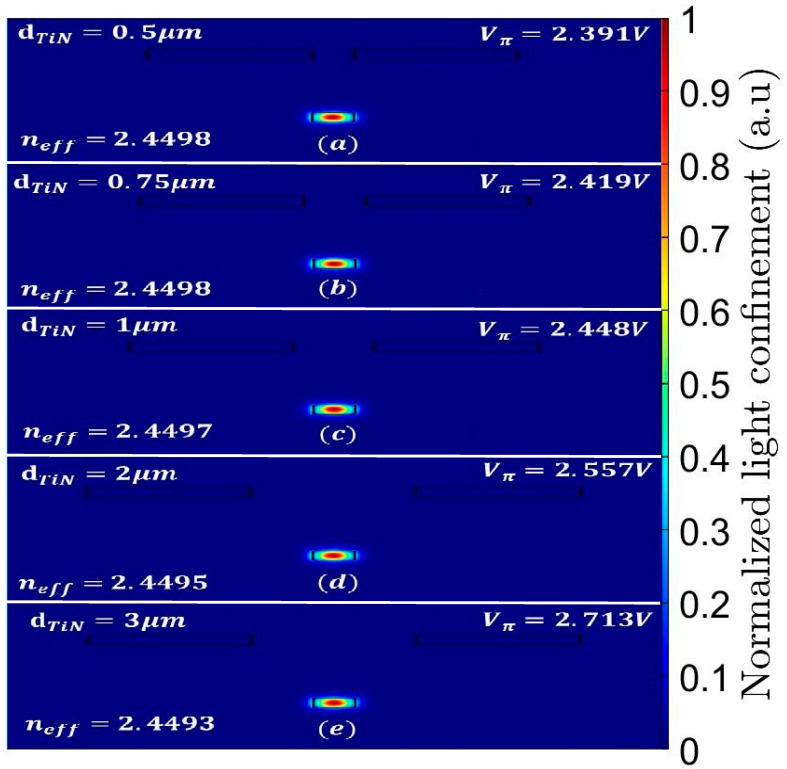
Solution of the fundamental TE mode at 1550 nm wavelength inside the Si strip: (**a**) d_TiN_ = 0.5 μm, (**b**) d_TiN_ = 0.75 μm, (**c**) d_TiN_ = 1 μm, (**d**) d_TiN_ = 2 μm, (**e**) d_TiN_ = 3 μm.

**Figure 4 sensors-23-08587-f004:**
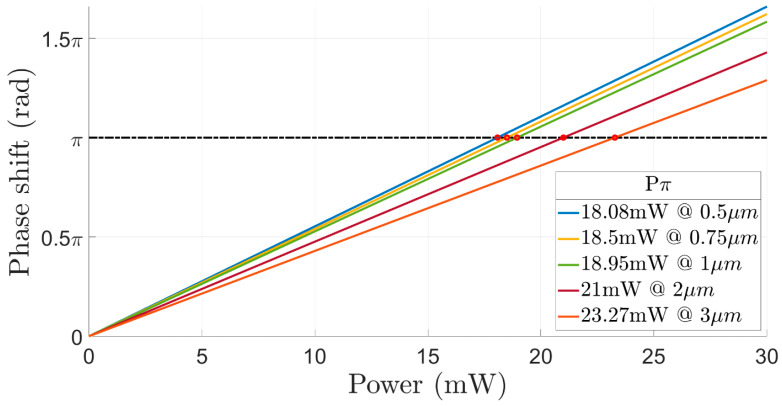
Phase shift as a function of power at different distances between the heaters.

**Figure 5 sensors-23-08587-f005:**
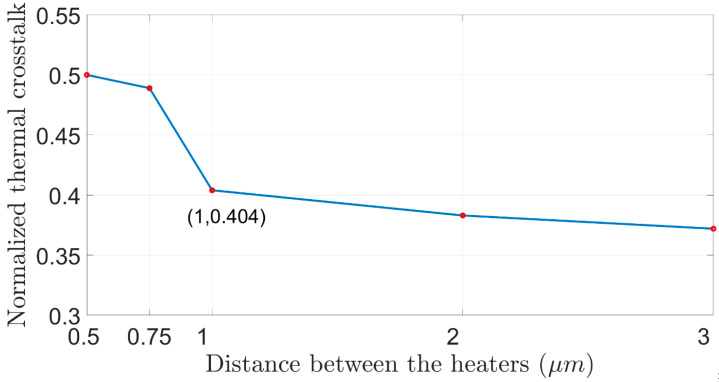
Normalized thermal crosstalk as a function of distance between the heaters.

**Figure 6 sensors-23-08587-f006:**
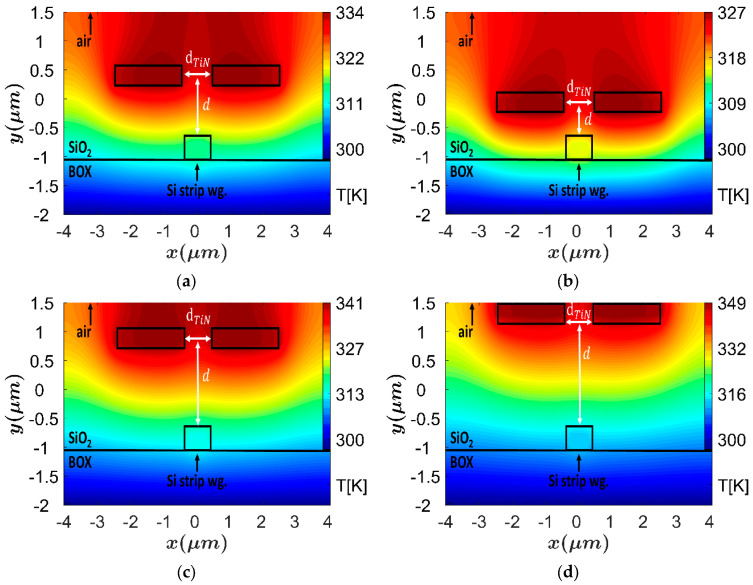
Heat distribution of the TOPS under different height distances between the heaters centered to the top of the Si strip wg: (**a**) d = 0.5 μm, (**b**) d = 1 μm, (**c**) d = 1.5 μm, (**d**) d = 2 μm.

**Figure 7 sensors-23-08587-f007:**
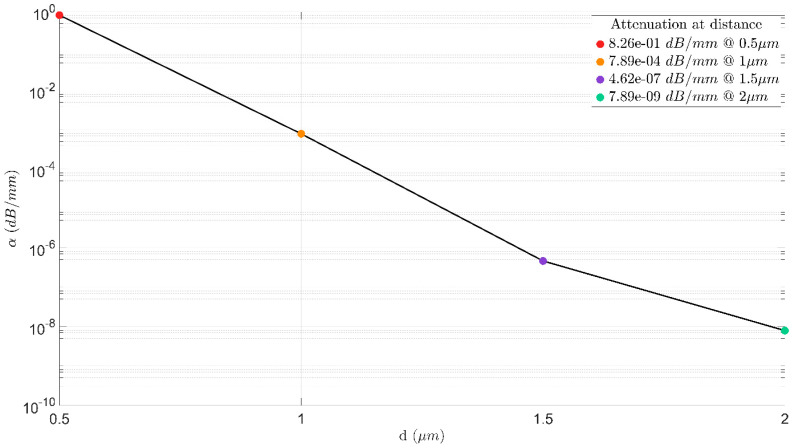
Attenuation as a function of parameter d.

**Figure 8 sensors-23-08587-f008:**
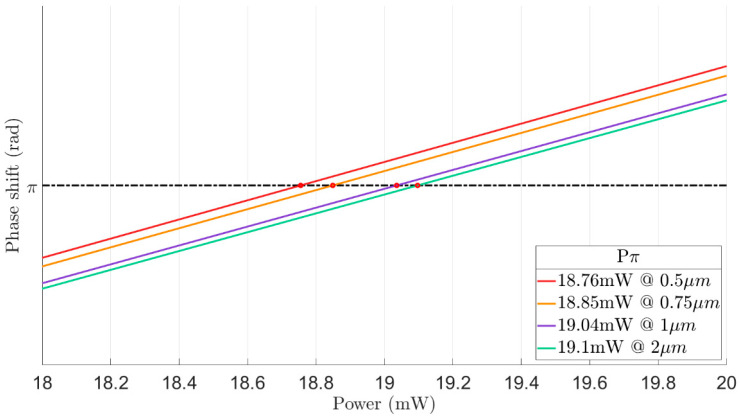
Total power consumption for π-phase shift as a function of parameter d. The results in Figure 8 correspond by color to the results in Figure 7.

**Figure 9 sensors-23-08587-f009:**
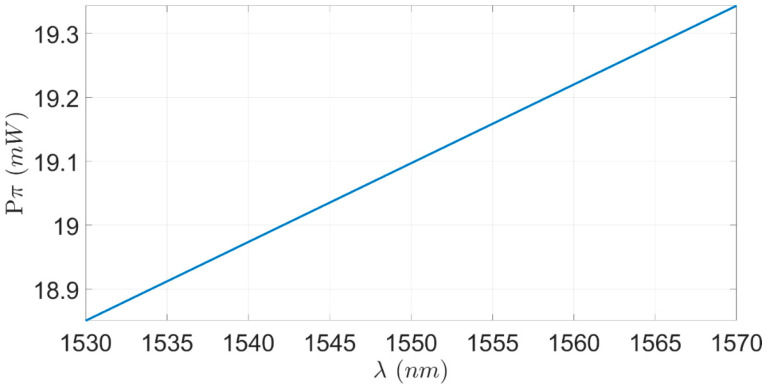
P_π_ as a function of the wavelengths over the C-band spectrum.

**Table 1 sensors-23-08587-t001:** A comparison between different types of TOPS.

TOPS Type	L (μm)	Losses (dB)	V_π_ (v)	P_π_ (mW)	τ (μs)	FOM P_π_ τ (mWμs)	Year of Publication
Doped Si (P) [7]	61.6	0.23	4.36	24.8	2.7	66.9	2014
Doped Si (N) [39]	320	0.08	36.4	19.1	2.471	47.2	2022
TiN [4]	320	<0.4	2.33	21.4	5.6	119.8	2018
Double TiN	1000	7.89 × 10^−4^	3.07	19.1	2.31	44.12	Our work

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
