# Peer review of "Optimizations of Double Titanium Nitride Thermo-Optic Phase-Shifter Heaters Using SOI Technology"

_sensors, 2023, doi:10.3390/s23208587_

Round 1

Reviewer 1 Report

1. Begin by explicitly stating the problem you are addressing and the specific objective of your research.

2. Include a brief summary of the methodology you employed to investigate the proposed double titanium nitride heaters model.

3. Quantitative Data Where possible, include quantitative data and results to provide a more concrete understanding of the findings.

4. Ensure consistency in terminology and acronyms throughout the abstract and manuscript.

5. Review the structure and flow of the abstract to ensure that it reads logically and succinctly.

 6. Conclude the abstract with a sentence or two discussing the broader implications of your research and its potential impact on the field.

Extensive editing of English language required. 

Author Response

Dear Reviewer,

Thank you for your mail from October 6th regarding the review results of manuscript id: Sensors-2650973: "Optimizations of double Titanium Nitride thermo-optic phase-shifter heaters using SOI technology". We have revised the manuscript while addressing all the comments made by the reviewers. We added new results and clarification text. Our detailed reply to the comments made by the reviewers can be seen below. I hope that in its revised form you may find the manuscript suitable for publication in Sensors.

Reviewer 1:

Begin by explicitly stating the problem you are addressing and the specific objective of your research.

Answer:

We thank the reviewers for their comment.

  • In this paper, we address the “Imbalance fabrication problem”, which is a difference between the Mach-Zehnder modulator ARMS due to manufacturing defects, and is being solved by the Thermo-Optic Phase-Shifter, in order to match the phase at the arms output. Most of the MZMs based TOPS today, have a single, usually directly attached TOPS unit made of Si, where in some cases it’s doped Si. As shown in the paper, the doped Si, is less power efficient (8mW for a pi-phase shift) and has substantial attenuation (0.23 dB).
  • The objective of our research is to purpose a new improved model, which uses less power consumption, (in our case, as low as 25% less power) while also featuring an outstandingly low attenuation coefficient (73e-4dB) a tiebreaker that allow implementation in long range communication systems. Such low attenuation is rarely seen nor being purposed since the heating unit is usually attached to the wg. A design which spikes the attenuation.

Reviewer 1:

Include a brief summary of the methodology you employed to investigate the proposed double titanium nitride heaters model.

Answer:

  • The use of Titanium nitride rose from the review of literature, specifically paper [8]: “Optimization of thermo-optic phase-shifter design and mitigation of thermal crosstalk on the SOI platform”. Where we have seen the use Titanium nitride, and it became our chosen material due to its low specific heat (598 J/kgK) versus (711 J/kgK) in Si.

  • The implementation of dual heaters design came as an idea, to spatially dissipate the heat more efficiently, and generally, purpose a new approach for heat dissipation, that will cover more of the wg. surface, while decrementing the power consumption.
  • Our investigation began with a simulation of heat dissipation in steady state; each simulation varied by the distance between the heaters. The results consisted of two graphs: The first one, a measure of phase shift of a propagating signal, with wavelength of 1550nm, as a function of heaters’ total power consumption, and the second respectively, a measure of the crosstalk as a function of distance between the heaters. The chosen distance was picked based on a tolerated crosstalk and a total power consumption which was substantial lower than previous works.

  • After chosing the fixed distance between the heaters, we wanted to check the trade-off between the power consumption and the attenuation coefficient. The closer the heaters are, the bigger the attenuation and vice versa. Getting to 2um, we get a substantial increase in power consumption which is closer to the literature, hence we picked the 1um distance, which had marginal increase in power which can be justified due to the incredibly lower attenuation.

  • At last, we did a profound investigation of the design behavior based on wavelength in the commonly used C band (due to its global minimum attenuation). We can see a roughly linear dependency between the power consumption and the propagating wavelength. With respect to the 1550nm wavelength, the increase and decrease respectively between the total power and the wavelength is marginal, and still is more power efficient by at least 22% percent.

Concluded results and comprehensive comparison can be viewed at table 2.

Reviewer 1:

Quantitative Data Where possible, include quantitative data and results to provide a more concrete understanding of the findings.

Answer:

Indeed, there is quite quantitative data in the findings such as in the phase-power consumption figure 4. And the profound investigation in figure 7. With that being said, we found it to be more concise to show the reader the optimized points of interest, and the entire data in the graph as a linear relations, which can be also be used in the future with the same design and different power consumption /crosstalk demand, etc.

Reviewer 1:

Ensure consistency in terminology and acronyms throughout the abstract and manuscript.

Answer:

  • Used terminology is abbreviated solely on the used design such as Silicon on Insulator (SOI) or used materials such as Silica (SiO2)

Reviewer 1:

Review the structure and flow of the abstract to ensure that it reads logically and succinctly.

Answer:

  • Abbreviated terms are fully mentioned once.

  • Key features and simulation results are written concisely.

Reviewer 1:

Conclude the abstract with a sentence or two discussing the broader implications of your research and its potential impact on the field.

Answer:

  • The purposed design is more power efficient by at least 22%, compared to previous work.

  • The attenuation is substantial smaller unlike other compared designs making it a potential candidate for usage in long range communication systems.

  • The usage of detached heaters with low specific heat, has been found to be superior both in power consumption and in attenuation, making the commonly used doped Si less appealing for future usage and should be prioritized instead of doping concentrations for future research. Also, the paper generally encourages towards the power advantage of the multiple heaters usage due to their larger surface cover.

Author Response

Dear Reviewer,

Thank you for your mail from October 6th regarding the review results of manuscript id: Sensors-2650973: "Optimizations of double Titanium Nitride thermo-optic phase-shifter heaters using SOI technology". We have revised the manuscript while addressing all the comments made by the reviewers. We added new results and clarification text. Our detailed reply to the comments made by the reviewers can be seen below. I hope that in its revised form you may find the manuscript suitable for publication in Sensors.

Authors have reported a design of thermo-optic phase shifter with double TiN heaters using a conventional silicon waveguide structure. Through numerical analysis they have optimized the separation between the two heaters and in between the heaters and the waveguide to find out the best possible configuration. Authors should address the below comments in detail.

Reviewer 2:

It is required to consider a single heater in the study to show the benefit of using two heater configurations. Authors should consider adding related plots in Fig.2 and 3 as well.

Answer:

We thank the reviewer for their comment. The measurement of a single heater is based on paper [8]: “Optimization of thermo-optic phase-shifter design and mitigation of thermal crosstalk on the SOI platform”. We’ve used its results solely for the comparison, as seen in table 1. The single heater design has no further contribution to be viewed by its heat dissipation, that hasn’t been already address in previous work  .

Reviewer 2:

Instead of showing Table.1, I recommend adding graphs to show the variation of attenuation and power with the separation between the waveguide and heaters.

Answer:

Thank you for the comment, we’ve replaced the table with a graph.

Reviewer 2:

In order to show the response time for thermo-optic phase shifter it is common to report the rising as well as falling time response (doi.org/10.1364/OE.18.018312, doi.org/10.1364/OE.461876). It should be included in the study as well as a comparison with the previously reported results.

Answer:

The rise/fall time is mentioned in Table 2.

Reviewer 2:

In line No. 34, authors should mention the TM mode as well. There is no hardbound rule that the MZI can only work with TE mode.

Answer:

Thank you for the comment, we’ve edited that part respectively.

Reviewer 2:

The definition of normalized thermal crosstalk is not clear. What are X and Y? Is it the temperature distribution? A clear description should be added.

Answer:

The surface in the XY plane, is being viewed as a matrix, and each point being summed over is a location on the matrix. The Ynm Component is the passive component surface, which in our case is the wg. and Xqr is the active components surface which is the heaters surface.

Reviewer 3 Report

The article discusses the optimization of double Titanium Nitride (TiN) thermo-optic phase shifter heaters for use in Mach-Zehnder modulators (MZM). The main findings and claims are:

1.      The article highlights the importance of thermo-optic phase shifters in Mach-Zehnder modulators to address the imbalance in signal paths. It states that current designs use a single heater, which requires more electrical power and increases costs.

While the article sets up the problem well, it lacks a clear statement of its research objectives and hypotheses. What precisely is the article aiming to optimize, and what are the research questions?

2.      The article describes the use of Titanium Nitride (TiN) heaters, their advantages over doped semiconductors, and the simulation tools used for thermal and optical analysis.

The article should provide more details about the simulation setup and the assumptions made. Additionally, it would be helpful to explain why TiN heaters were chosen over other materials and how they were integrated into the system.

3.      The article presents results for different distances between the TiN heaters, showing their impact on thermal crosstalk, power consumption, and optical losses. It also discusses the sensitivity of the phase shift to laser wavelength drift.

The results are presented clearly, but the article could benefit from a more comprehensive analysis of the findings. Are there any trade-offs between power consumption, thermal crosstalk, and other factors? How do these results compare to existing solutions?

4.      The article identifies an optimal design with specific parameters, such as the distance between heaters and the distance from heaters to the waveguide. It claims that this design minimizes power consumption while maintaining low losses.

The article should thoroughly justify why this design is optimal and how these parameters were determined. Additionally, it should discuss potential limitations or constraints associated with this design.

5.      The article compares the proposed design with other existing methods, highlighting its advantages in terms of power consumption, thermal crosstalk, and rise time.

The comparison is valid, but the article should also discuss potential drawbacks or limitations of the proposed design compared to existing solutions. Are there any trade-offs that need to be considered?

6.      The article concludes that the new TOPS design offers improved energy efficiency and can be integrated into high-speed systems with large RF lines, reducing transmitter costs.

The conclusion is somewhat repetitive and could benefit from summarizing the essential findings and their implications more concisely.

7.      The article could benefit from raising questions or suggesting directions for future research. Are there any unresolved issues or challenges related to this technology?

The article presents an exciting optimization study of double TiN heaters for thermo-optic phase shifters. However, it would be enhanced by providing more context on the research objectives, a deeper analysis of the results, and a discussion of potential limitations and future research directions.

Author Response

Dear Reviewer,

Thank you for your mail from October 6th regarding the review results of manuscript id: Sensors-2650973: "Optimizations of double Titanium Nitride thermo-optic phase-shifter heaters using SOI technology". We have revised the manuscript while addressing all the comments made by the reviewers. We added new results and clarification text. Our detailed reply to the comments made by the reviewers can be seen below. I hope that in its revised form you may find the manuscript suitable for publication in Sensors.

The article discusses the optimization of double Titanium Nitride (TiN) thermo-optic phase shifter heaters for use in Mach-Zehnder modulators (MZM). The main findings and claims are:

Reviewer 3:

The article highlights the importance of thermo-optic phase shifters in Mach-Zehnder modulators to address the imbalance in signal paths. It states that current designs use a single heater, which requires more electrical power and increases costs. While the article sets up the problem well, it lacks a clear statement of its research objectives and hypotheses. What precisely is the article aiming to optimize, and what are the research questions?

Answer:

We thank the reviewer for their comment.

The article optimizes the design such that it will have substantial low attenuation, tolerated crosstalk and lower total power consumption, all with respect to previous work (mentions in table 2).

The optimized degrees of freedom are the distance between the heaters and the distance between the center of mass of the heaters and the wg. all being simulated at wavelength of 1550nm.

Reviewer 3:

The article describes the use of Titanium Nitride (TiN) heaters, their advantages over doped semiconductors, and the simulation tools used for thermal and optical analysis. The article should provide more details about the simulation setup and the assumptions made. Additionally, it would be helpful to explain why TiN heaters were chosen over other materials and how they were integrated into the system.

Answer:

In the introduction part, it’s explained that the Titanium Nitride was chosen due it’s low specific heat (598 J/kgK) versus (711 J/kgK) in Si. Their advantage over doped semi conductors is both mentioned in the conclusions, as well as in table 1. Where it can be clearly seen that it is both more power efficient by at least 22%, and attenuation is substantial smaller unlike other compared designs making it a potential candidate for usage in long range communication systems.

Reviewer 3:

The article presents results for different distances between the TiN heaters, showing their impact on thermal crosstalk, power consumption, and optical losses. It also discusses the sensitivity of the phase shift to laser wavelength drift. The results are presented clearly, but the article could benefit from a more comprehensive analysis of the findings. Are there any trade-offs between power consumption, thermal crosstalk, and other factors? How do these results compare to existing solutions?

Answer:

The trade-off between the power consumption and the thermal crosstalk can be viewed in fig. 4. And 5. Respectively, the description between them describe profoundly the optimized decision for the distance of 1 um.

Reviewer 3:

The article identifies an optimal design with specific parameters, such as the distance between heaters and the distance from heaters to the waveguide. It claims that this design minimizes power consumption while maintaining low losses. The article should thoroughly justify why this design is optimal and how these parameters were determined. Additionally, it should discuss potential limitations or constraints associated with this design.

Answer:

By comparison in table 1. We’ve shown that total power consumption, FOM and attenuation we’ve achieved far superior results compared to recent work.

Reviewer 3:

The article compares the proposed design with other existing methods, highlighting its advantages in terms of power consumption, thermal crosstalk, and rise time. The comparison is valid, but the article should also discuss potential drawbacks or limitations of the proposed design compared to existing solutions. Are there any trade-offs that need to be considered?

Answer:

Table 1. shows that with the usage of only a single TiN heater, which has it’s own drawbacks such as much lower rise/fall time (5.6 us)

The emphasis on the parallel heaters distributes evenly the power between them, hence, the rise time for each unit is much smaller.

Other than cost production, the suggested design is more efficient in the commonly compared aspects in table 1.

Our intention was to show the design solely in the C band where it’s commonly used today in optical communications due to the global minimum attenuation in this band (specifically at 1550nm).

Reviewer 3:

The article concludes that the new TOPS design offers improved energy efficiency and can be integrated into high-speed systems with large RF lines, reducing transmitter costs. The conclusion is somewhat repetitive and could benefit from summarizing the essential findings and their implications more concisely.

Answer:

Due to the optimized degrees of freedom, we found it to be more convenient for the reader to view our optimized results summed up altogether in the conclusions.

Reviewer 3:

The article could benefit from raising questions or suggesting directions for future research. Are there any unresolved issues or challenges related to this technology?

Answer:

There has been usage of MZM in recent photonic computers / quantum computers for phase corrections. With that being said, the design in the paper is for any kind of general use in optical communications.

Reviewer 3:

The article presents an exciting optimization study of double TiN heaters for thermo-optic phase shifters. However, it would be enhanced by providing more context on the research objectives, a deeper analysis of the results, and a discussion of potential limitations and future research directions.

Answer:

Thank you for the comment, please view the answer from drawback /limitations questions.

Round 2

Reviewer 1 Report

Accepted. 

Minor editing of English language required

Reviewer 2 Report

The authors' responses to most of the quires are satisfactory. It can be considered for publication now.

Reviewer 3 Report

The authors have addressed the comments. It may be accepted for publication.